# Mediterranean Diet, Polyphenols, and Neuroprotection: Mechanistic Insights into Resveratrol and Oleuropein

**DOI:** 10.3390/nu17243929

**Published:** 2025-12-16

**Authors:** Mónika Fekete, Tamás Jarecsny, Andrea Lehoczki, Dávid Major, Vince Fazekas-Pongor, Tamás Csípő, Ágnes Lipécz, Ágnes Szappanos, Eszter Melinda Pázmándi, Péter Varga, János Tamás Varga

**Affiliations:** 1Institute of Preventive Medicine and Public Health, Semmelweis University, 1089 Budapest, Hungary; ceglediandi@freemail.hu (A.L.); major.david@semmelweis.hu (D.M.); pongor.vince@semmelweis.hu (V.F.-P.); csipo.tamas@semmelweis.hu (T.C.); lipecz.agnes@semmelweis.hu (Á.L.); varga.peter@semmelweis.hu (P.V.); 2Fodor Center for Prevention and Healthy Aging, Semmelweis University, 1089 Budapest, Hungary; 3Doctoral College, Semmelweis University, 1085 Budapest, Hungary; tjarecs@gmail.com (T.J.); pazmandi.eszter.melinda@semmelweis.hu (E.M.P.); 4Department of Neurology and Stroke, Saint John’s Central Hospital of North Buda, 1125 Budapest, Hungary; 5Department of Rheumatology and Clinical Immunology, Semmelweis University, 1023 Budapest, Hungary; drszappanos@gmail.com; 6Heart and Vascular Center, Semmelweis University, 1122 Budapest, Hungary; 7András Pető Faculty, Semmelweis University, 1125 Budapest, Hungary; 8Department of Pulmonology, Semmelweis University, 1083 Budapest, Hungary

**Keywords:** Mediterranean diet, neurodegeneration, Alzheimer’s disease, Parkinson’s disease, mild cognitive impairment, polyphenols, resveratrol, oleuropein, oxidative stress, inflammation, gut–brain axis

## Abstract

**Background:** Neurodegenerative diseases, including Alzheimer’s disease and Parkinson’s disease, are among the leading causes of disability and mortality worldwide. Dietary patterns have emerged as modifiable risk factors that may influence disease onset and progression. The Mediterranean diet (MedDiet), rich in fruits, vegetables, whole grains, legumes, fish, and extra virgin olive oil, has been consistently associated with better cognitive outcomes and reduced risk of neurodegeneration. **Aim:** This narrative review summarizes current evidence on the role of the MedDiet in slowing the progression of neurodegenerative diseases, with a particular focus on polyphenols such as resveratrol and oleuropein as key bioactive mediators. **Methods:** We synthesized findings from epidemiological studies, clinical trials, and mechanistic research to provide an integrated overview of how adherence to the MedDiet and its polyphenol components affects neurodegenerative disease trajectories. **Results:** Epidemiological studies suggest that higher MedDiet adherence is associated with slower cognitive decline, reduced conversion from mild cognitive impairment to Alzheimer’s disease, and better motor and non-motor outcomes in Parkinson’s disease. Mechanistically, the MedDiet modulates oxidative stress, neuroinflammation, mitochondrial function, vascular health, and the gut–brain axis. Polyphenols such as resveratrol and oleuropein exert neuroprotective effects through antioxidant activity, modulation of amyloid aggregation, mitochondrial biogenesis, and activation of signaling pathways (e.g., SIRT1). Clinical studies, although limited, indicate beneficial effects of polyphenol-rich interventions on cognitive and metabolic biomarkers. **Conclusions:** Current evidence supports the Mediterranean diet as a promising dietary strategy to slow the progression of neurodegenerative diseases. Polyphenols, including resveratrol and oleuropein, may play a role in mediating these effects. Further well-designed, long-term clinical trials are needed to establish causal relationships, optimize dosage, and explore biomarker-driven personalized nutrition approaches.

## 1. Introduction

Neurodegenerative diseases, including Alzheimer’s disease and Parkinson’s disease, are among the most pressing public health challenges worldwide [1,2,3,4,5]. Their prevalence increases steeply with age, and they contribute substantially to disability, loss of independence, and premature mortality [2]. Although pharmacological therapies continue to advance, current treatments have only modest effects on slowing disease progression, and none provide meaningful disease modification [6,7,8,9,10]. As a result, there is growing emphasis on identifying preventive and adjunctive strategies that may preserve cognitive function and delay neurodegenerative processes across the lifespan [11,12,13,14].

Dietary patterns have emerged as one of the most influential and modifiable determinants of brain aging [3,15,16,17,18,19,20,21,22,23,24,25]. Among them, the Mediterranean diet (MedDiet) has received particular attention [3,15,17,19,20,23,26,27]. Characterized by high intake of fruits, vegetables, legumes, whole grains, nuts, fish, and extra virgin olive oil; moderate consumption of dairy products and wine; and low intake of red and processed meats, the MedDiet represents a nutrient-dense, anti-inflammatory, and metabolically favorable dietary pattern [14,27,28,29,30,31,32,33,34,35,36]. Numerous epidemiological and interventional studies have linked the MedDiet to improved cardiovascular and metabolic health and to better cognitive performance in older adults [15,27,29,37,38,39,40,41,42].

Within the MedDiet, polyphenols have been proposed as key contributors to neuroprotection [13]. Among these, resveratrol—abundant in grapes, berries and red wine [43,44,45]—and oleuropein—characteristic of olives and olive oil [46,47]—have attracted particular interest. Experimental and clinical studies suggest that these compounds exert antioxidant [48,49,50,51,52,53], anti-inflammatory [43,54,55,56,57], and neuroprotective effects, influence mitochondrial function [58,59,60,61,62], and modulate protein aggregation pathways relevant to Alzheimer’s and Parkinson’s disease [63,64,65]. They also activate key signaling cascades involved in cellular stress resistance and brain aging, including SIRT1 [66,67,68,69,70,71].

The aim of this narrative review is to summarize current evidence on the Mediterranean diet and the progression of neurodegenerative diseases, with particular focus on polyphenols—especially resveratrol and oleuropein—as potential mediators of these associations. By integrating epidemiological findings, clinical trial data, and mechanistic insights, the review highlights how the MedDiet may influence neurodegenerative trajectories and discusses key areas where further research is needed to refine preventive and therapeutic applications.

## 2. Methods

### 2.1. Literature Search

The literature search was conducted in the PubMed, Scopus, and Web of Science databases, covering the period from January 2010 to August 2025. The following keywords and their combinations were used: “Mediterranean diet,” “neurodegenerative diseases,” “Alzheimer’s disease,” “Parkinson’s disease,” “cognitive decline,” “polyphenols,” “resveratrol,” “oleuropein,” “oxidative stress,” and “neuroprotection.” This search initially yielded 5732 records (PubMed: 1856; Scopus: 1944; Web of Science: 1932). After removing 1324 duplicates, 4408 records remained for screening. Titles and abstracts were screened for relevance, leading to the exclusion of 4212 studies that did not meet the inclusion criteria. The full texts of 196 studies were assessed for eligibility, and 70 studies were ultimately included in the qualitative synthesis. The search was further supplemented by manual screening of references cited in relevant reviews and clinical studies.

### 2.2. Inclusion Criteria

Studies were considered eligible if they met the following criteria:Population: Human studies related to neurodegenerative diseases or cognitive decline.Intervention/exposure: Examination of the Mediterranean diet as a whole or its key bioactive components [e.g., resveratrol, oleuropein, other polyphenols].Outcomes: Incidence or progression of neurodegenerative diseases, changes in cognitive function, or biomarkers [e.g., oxidative stress, inflammatory markers, amyloid and tau pathology].Study types: Randomized controlled trials (RCTs), prospective and retrospective cohort studies, and cross-sectional studies, but not experimental [in vivo or in vitro] research.

### 2.3. Exclusion Criteria

The following were excluded:Publications without original data [editorials, letters, conference abstracts].Studies not related to neurodegenerative diseases or cognitive outcomes.Studies investigating isolated nutrients outside the context of the Mediterranean diet.Animal or other preclinical studies.

### 2.4. Data Extraction and Synthesis

From each eligible study, we extracted information on authorship, publication year, study population, sample size, exposure or intervention, outcomes, and key results. Evidence was synthesized narratively and grouped into the following themes:Epidemiological associations between the Mediterranean diet and cognitive decline or neurodegenerative diseases.Findings from clinical trials of Mediterranean diet-based interventions.Mechanistic insights focusing on polyphenols, particularly resveratrol and oleuropein.Remaining knowledge gaps and directions for future research.

### 2.5. Quality Assessment and Limitations

As a narrative review, no formal quality appraisal tool was applied. However, when interpreting the literature, we prioritized studies with strong methodological design, adequate sample size, and longer follow-up. Potential biases, heterogeneity across studies, and publication bias were taken into consideration.

### 2.6. Aim of the Review

The aim of this narrative review is to synthesize current evidence on how the Mediterranean diet may influence the progression of neurodegenerative diseases, with particular emphasis on the potential contributions of polyphenols such as resveratrol and oleuropein.

It is important to emphasize that focusing on resveratrol and oleuropein does not imply that these compounds are the sole or principal drivers of the Mediterranean diet’s neuroprotective effects. Dietary intake levels of both molecules are generally very low, and the concentrations used in mechanistic studies—particularly in vitro and animal models—often exceed those achievable through habitual Mediterranean dietary patterns. Their inclusion in this review reflects their status as two of the most extensively investigated bioactive constituents of the diet, rather than exclusive mediators of its benefits. Accordingly, the relevance of these mechanistic pathways to human physiology is limited by substantial dose discrepancies, and the evidence presented should be interpreted as supporting biological plausibility rather than direct proof of efficacy at dietary exposure levels. The overall neuroprotective potential of the Mediterranean diet is more likely to stem from synergistic interactions among multiple dietary components—including other olive-derived phenolics, monounsaturated fatty acids, antioxidant- and fiber-rich plant foods, and gut–brain axis-mediated metabolic effects. Therefore, the discussion of resveratrol and oleuropein serves primarily as an illustrative framework for understanding specific mechanistic pathways, without overstating their isolated importance.

## 3. Mechanistic Basis: How the Mediterranean Diet May Slow Neurodegeneration

The neuroprotective effects of the MedDiet arise from a constellation of biological mechanisms supported by its nutrient composition and bioactive compounds. These mechanisms provide a rationale for epidemiological observations linking the MedDiet to slower cognitive decline and reduced neurodegenerative risk. Table 1 provides a comparative overview of key dietary patterns, highlighting the nutrients and bioactive compounds that may influence neurodegenerative processes.

### 3.1. Anti-Inflammatory Effects

Chronic low-grade inflammation is a central driver of neurodegenerative diseases [72,73,74,75], and several components of the MedDiet act to counter this process [23,76,77,78,79]. The diet’s high content of omega-3 fatty acids, olive oil phenolics, polyphenols, carotenoids, and dietary fiber contributes to lower circulating levels of proinflammatory cytokines such as interleukin-6 (IL-6) and tumor necrosis factor-alpha (TNF-α). Additional micronutrients—including zinc, magnesium, and calcium—support immune regulation. Extra virgin olive oil provides phenolic compounds such as oleocanthal, oleacein, and oleic acid, which inhibit nuclear factor kappa-light-chain-enhancer of activated B cells (NF-κB)-related inflammatory pathways and platelet activation. By engaging these complementary pathways, the MedDiet reduces systemic inflammation and may thereby mitigate neuroinflammatory cascades implicated in cognitive decline and neurodegeneration [80].

### 3.2. Antioxidant Effects and Protection Against Protein Misfolding

Oxidative stress is a key mediator of neuronal injury, protein misfolding, and synaptic dysfunction in Alzheimer’s and Parkinson’s disease [81,82,83,84]. The MedDiet’s rich supply of polyphenols, vitamins E and C, and carotenoids enhances cellular antioxidant capacity and reduces reactive oxygen species (ROS) [85,86,87,88]. Polyphenols play a particularly important role, as their antioxidant and anti-inflammatory properties counteract protein aggregation processes linked to the formation of amyloid-β and α-synuclein deposits, which are central features of Alzheimer’s and Parkinson’s disease pathophysiology. Experimental models have demonstrated that resveratrol promotes amyloid-β degradation through proteasomal and autophagosomal pathways, increases the activity of degrading enzymes such as neprilysin, and inhibits the pathological hyperphosphorylation of tau protein. Furthermore, resveratrol directly interferes with Aβ aggregation and facilitates the formation of non-toxic conformations. Similarly, quercetin enhances AMPK activity and reduces tau hyperphosphorylation, while other flavonoids—including anthocyanins and caffeic acid derivatives—exert neuroprotective effects by attenuating oxidative stress and protein aggregation. Collectively, these mechanisms enhance neuronal resilience against toxic protein accumulation and ROS-induced damage, thereby highlighting the antioxidant components of the MedDiet as key contributors to the slowing of neurodegenerative processes [89].

### 3.3. Support of Mitochondrial Function

Mitochondrial dysfunction is a recognized early event in the pathogenesis of Alzheimer’s and Parkinson’s disease, contributing to impaired energy metabolism, increased oxidative stress, and heightened neuronal vulnerability [90,91,92,93,94,95,96,97,98,99]. Bioactive components of the Mediterranean diet—including resveratrol and major polyphenols from extra virgin olive oil (EVOO), such as hydroxytyrosol, oleuropein, and oleocanthal—play a crucial role in maintaining mitochondrial health [76,100,101]. These compounds activate key energy-regulating pathways, including AMP-activated protein kinase (AMPK), sirtuin-1 (SIRT1), and the mechanistic target of rapamycin (mTOR), which converge on transcriptional regulators of mitochondrial biogenesis such as peroxisome proliferator-activated receptor gamma coactivator-1 alpha (PGC-1α), nuclear respiratory factor-1 (NRF1), and mitochondrial transcription factor A (TFAM) [66,102,103,104,105,106]. Through these pathways, they stimulate the formation of new mitochondria, enhance ATP production, and help maintain mitochondrial DNA stability [59,60,61,62,67,107,108,109]. Experimental studies consistently show that EVOO polyphenols increase ATP levels, improve respiratory chain complex activity, and restore oxidative phosphorylation efficiency. In parallel, EVOO-derived phenolics reduce mitochondrial ROS generation, strengthen endogenous antioxidant defenses, and activate the Nrf2-dependent vitagene network, leading to increased expression of cytoprotective enzymes such as superoxide dismutase and catalase. By also suppressing NF-κB-mediated inflammatory signaling, they blunt both oxidative and inflammatory stress in cells of the neurovascular unit [43,58,66,105,110,111,112,113,114]—two interlinked drivers of neurodegenerative pathology. Beyond these effects, EVOO polyphenols promote mitophagy, stabilize mitochondrial membrane structure and fluidity, and support a healthy balance between mitochondrial fusion and fission. Together, these actions counteract age- and metabolism-related mitochondrial dysfunction. Because neurons depend heavily on efficient mitochondrial function, these mechanisms are likely to contribute substantially to the neuroprotective effects of the Mediterranean diet [115]. It is important to note that certain mechanistic elements presented in this section overlap with those discussed in Section 3.4, as mitochondrial and vascular processes are biologically tightly interconnected.

### 3.4. Improvement of Vascular and Metabolic Health

Vascular dysfunction [116,117,118,119,120,121] and metabolic impairment [23,72,122,123] are major contributors to Alzheimer’s disease and cognitive decline [124,125,126,127,128], influencing cerebral perfusion, blood–brain barrier integrity, insulin signaling, and the accumulation of amyloid and tau pathology. The MedDiet favorably modulates glucose and insulin metabolism [23,35,87], enhances endothelial and neurovascular function [129,130,131,132,133,134], and reduces the risk of atherosclerosis. These improvements support cerebral blood flow and neuronal energy supply and mitigate vascular contributions to cognitive impairment and dementia. The MedDiet’s emphasis on plant-based foods—fruits, vegetables, legumes, whole grains, and nuts—provides antioxidants, anti-inflammatory nutrients, and fiber that collectively reduce insulin resistance, endothelial dysfunction, and cardiovascular risk. Polyphenols play a particularly important role: compounds such as resveratrol, quercetin, catechins, and ellagic acid help maintain vascular homeostasis by increasing endothelial nitric oxide availability, reducing adhesion molecule expression (ICAM-1, VCAM-1), and lowering inflammatory cytokines, including TNF-α and IL-6 [61,110,111,112,114,129,135,136,137,138,139,140,141,142,143,144,145,146]. Together, these mechanisms slow vascular aging by limiting oxidative stress, reducing endothelial senescence and telomere-related damage, and dampening vascular inflammation. By preserving endothelial function, improving metabolic control, and lowering cardiometabolic risk factors, the MedDiet supports cardiovascular health and, indirectly, contributes to reducing neurodegenerative risk and progression [25,147].

Functional brain connectivity is increasingly recognized as a key mechanistic component of Alzheimer’s disease and age-related cognitive decline [148,149,150,151,152,153,154]. Neurodegeneration disrupts large-scale neural networks—particularly the default mode, frontoparietal, and hippocampal networks—leading to impaired synchronization between regions that support memory, executive function, and attention. Resting-state fMRI studies consistently show reduced connectivity within the default mode network and altered hippocampal–prefrontal coupling even in early or preclinical AD stages [148,149,151,152,153]. Emerging evidence suggests that dietary patterns such as the Mediterranean diet may influence these network-level alterations [155,156]. Interventions enriched with polyphenols have been shown to enhance hippocampal functional connectivity [157] and improve cerebrovascular responsiveness and neurovascular coupling [51]. These findings raise the possibility that diet-induced modulation of neurovascular and metabolic pathways may partly restore or preserve functional network integrity, offering a complementary mechanism through which the Mediterranean diet could slow cognitive decline.

### 3.5. Gut–Brain Axis and Microbiome Modulation

The Mediterranean diet, rich in fiber, prebiotic compounds, and polyphenols, promotes a favorable gut microbial profile characterized by greater microbial diversity and increased production of short-chain fatty acids (SCFAs), especially butyrate [77,158]. These metabolites exert systemic effects that extend to the central nervous system by strengthening blood–brain barrier function, modulating innate and adaptive immune responses, and supporting neuronal energy metabolism. Butyrate and propionate, in particular, reduce inflammatory signaling, promote microglial homeostasis, and enhance mitochondrial efficiency. The MedDiet is associated with a microbiota composition enriched in beneficial butyrate-producing species such as *Faecalibacterium prausnitzii*, *Eubacterium rectale*, and *Roseburia* spp., while reducing proinflammatory or dysbiotic taxa [77,158,159,160,161,162,163,164]. Functional microbial outputs are equally important: SCFAs generated from the fermentation of dietary fibers and complex carbohydrates enhance regulatory T cell activity, lower systemic low-grade inflammation, and attenuate neuroinflammatory processes that contribute to neurodegenerative progression. Polyphenols and other plant-derived compounds further shape microbiome composition and activity. Phenolic constituents of olive oil, nuts, and red wine are metabolized by gut bacteria into bioactive derivatives with antioxidant and neuromodulatory effects. These metabolites promote the growth of *Bifidobacterium* and *Akkermansia* species and inhibit opportunistic pathogens such as *Ruminococcus gnavus*, which has been linked to increased gut permeability, metabolic endotoxemia, and systemic inflammation [77]. Overall, the MedDiet exerts a dual influence on the gut–brain axis: it shifts microbiome composition toward anti-inflammatory, SCFA-producing communities and increases the generation of metabolites that support neuronal health. These interrelated effects contribute to improved neuroprotection, preserved cognitive function, and a slower trajectory of neurodegenerative processes. It is important to note that much of the mechanistic evidence comes from in vitro or animal models, where the concentrations of polyphenols—particularly resveratrol and oleuropein—often exceed levels achievable through a typical Mediterranean diet. Direct biomarker-level evidence from human studies is limited, and daily dietary intake is far lower than experimental doses. Therefore, these mechanisms should be interpreted as providing biological plausibility rather than direct proof of effects at physiologically relevant dietary levels. Taken together, these mechanisms highlight the central role of the gut–brain axis in mediating the neuroprotective effects of the Mediterranean diet, and Figure 1 summarizes these interconnected pathways.

## 4. Polyphenols as Key Mediators

A defining feature of the Mediterranean diet is its high content of polyphenol-rich foods. Polyphenols influence several biological pathways relevant to neurodegeneration, including oxidative stress responses, inflammatory signaling, mitochondrial function, and protein aggregation dynamics. Below, we summarize the most extensively studied representatives.

### 4.1. Resveratrol

Resveratrol, a stilbenoid polyphenol abundant in grapes, berries and red wine, is among the most widely studied natural compounds in the context of Alzheimer’s disease [165,166,167,168,169,170,171,172,173]. Its neuroprotective actions are mediated through multiple interconnected mechanisms [174,175,176,177,178,179,180]. Its neuroprotective effects are mediated through multiple mechanisms: it reduces ROS through antioxidant and redox-regulating activity, restores glutathione levels, and activates antioxidant enzymes; SIRT1 activation inhibits NF-κB signaling, attenuates microglial overactivation, and supports neuronal survival; additionally, it modulates the AMPK/PGC-1α/mTOR pathways, enhances mitochondrial biogenesis, and optimizes ATP production. Resveratrol also reduces amyloid-β aggregation, mitigates tau hyperphosphorylation, regulates Cu^2+^, Zn^2+^, and Fe^2+^ homeostasis, and stimulates the expression of neurotrophic factors such as BDNF, NGF, and NT-3. Its anti-inflammatory effects are mediated through inhibition of MAPK and STAT1/3 signaling and suppression of iNOS and COX-2, while additional mechanisms include the induction of autophagy, activation of the Nrf2/HO-1/NQO1 pathway, and reduction in cholinesterase activity [71].

In cellular models, resveratrol attenuates Aβ1-42-induced mitochondrial damage and oxidative stress, promotes PGC-1α deacetylation via AMPK-dependent mechanisms, and activates the SIRT1 pathway, which inhibits NF-κB signaling, reduces proinflammatory cytokine (TNF-α, IL-1β, IL-6) production, and protects against microglial overactivation [181,182]. In animal studies, resveratrol has demonstrated multiple neuroprotective effects, including reduction in Aβ accumulation, decreased lipid peroxidation, enhanced expression of antioxidant enzymes, and improvement of spatial memory [183,184,185]. Furthermore, it stimulates the expression of neurotrophic factors in the nervous system, such as BDNF, NGF, and NT-3, promoting neurogenesis, neuronal survival, and cognitive function [186,187].

Although clinical studies have been limited by small sample sizes and short durations, they suggest that resveratrol can cross the blood–brain barrier and favorably modulate oxidative stress and inflammatory biomarkers, while potentially reducing Aβ accumulation [188,189,190]. Some studies, however, reported adverse effects such as weight loss, nausea, or diarrhea. Confirmation of long-term cognitive benefits requires further investigation. Due to its low bioavailability, pharmacokinetics, drug interactions, and formulation are critical factors for the clinical efficacy of resveratrol.

### 4.2. Olive Oil Polyphenols and Their Neuroprotective Effects

The extra virgin olive oil (EVOO) contains several potent phenolic compounds, most notably oleuropein (OLE), hydroxytyrosol (HT), and oleocanthal (OLC) [191]. These molecules exhibit strong antioxidant and anti-inflammatory properties and influence multiple pathways implicated in neurodegeneration [192,193]. Preclinical studies show that HT and OLE aglycone can cross the blood–brain barrier, while OLC enhances Aβ clearance and reduces neurotoxic aggregation [11,194,195].

Mechanistically, EVOO polyphenols activate the Nrf2/ARE antioxidant signaling pathway, increase expression of neurotrophic factors (BDNF, NGF), inhibit inflammatory mediators (NF-κB, iNOS, COX-2), attenuate mitochondrial dysfunction, and prevent apoptosis in oxidative stress-induced injury models [64]. These mechanisms parallel, and in some cases complement, those observed for resveratrol.

In vitro and in vivo studies across models of Alzheimer’s disease, Parkinson’s disease, Huntington’s disease, and amyotrophic lateral sclerosis consistently demonstrate reduced oxidative stress, less neuronal damage, and decreased inflammation associated with these compounds [64]. Clinical data—including findings from PREDIMED [196] and the MICOIL pilot study [197]—suggest that high-polyphenol EVOO improves cognitive performance, reduces blood–brain barrier permeability, lowers neurotoxic Aβ levels, and may slow cognitive decline. These results support the concept that regular EVOO consumption meaningfully contributes to the neuroprotective profile of the Mediterranean diet.

### 4.3. Other Polyphenols and Bioactive Compounds in the Mediterranean Diet

Beyond olive oil and resveratrol, the Mediterranean diet incorporates a wide array of other polyphenol-rich foods, including citrus fruits, berries, pomegranate, grapes, spices containing curcumin, and green tea rich in catechins [198,199]. These compounds exhibit antioxidant and anti-inflammatory effects and can modulate microglial activation, thereby reducing neuroinflammatory processes.

Many polyphenols also support synaptic function and cognitive resilience. Preclinical studies indicate that flavonoids, catechins, and curcumin enhance neuronal antioxidant defenses—often via Nrf2/ARE pathway activation—and increase levels of neurotrophic factors such as BDNF and NGF. These mechanisms promote synaptic plasticity, neuronal survival, and regeneration [200].

While clinical evidence in humans is still emerging, epidemiological data and animal experiments consistently support the neuroprotective potential of these bioactive compounds. Long-term consumption of polyphenol-rich foods within the framework of the Mediterranean diet may therefore contribute to healthy cognitive aging and reduce the risk of neurodegenerative diseases such as Alzheimer’s and Parkinson’s disease [201].

## 5. Mediterranean Diet and Neurodegeneration: Epidemiological Evidence

### 5.1. Cognitive Decline and Global Cognitive Performance

The relationship between the Mediterranean diet and cognitive function has been investigated in numerous epidemiological and interventional studies over the past decades [202,203,204,205,206,207,208,209,210,211,212]. Several prospective cohort studies have examined the associations between dietary patterns and the risk of dementia, mild cognitive impairment, or general cognitive decline in adults of various ages and sexes [210,211,212]. The results are mixed: some studies did not find a significant association between adherence to the Mediterranean diet and the risk of cognitive decline [210,213], whereas others reported beneficial effects, particularly for memory and global cognition [208,209].

Studies using a metabolomics approach, which assessed adherence to the Mediterranean diet based on biomarkers, also support its potential neuroprotective effects [214]. In cohort studies, higher adherence to the Mediterranean diet was especially associated with a reduced risk of cognitive decline in older women [215,216,217]. Clinical interventions, such as the PREDIMED and PREDIMED-NAVARRA trials, have demonstrated in randomized controlled settings that supplementing the Mediterranean diet with extra virgin olive oil or nuts significantly improves memory, executive functions, and overall cognitive performance compared to control groups [196,218]. Similarly, studies evaluating Mediterranean diet adherence have reported beneficial effects on memory, language, and visuospatial abilities, as well as mental health and quality of life [31,208,219].

Overall, current evidence suggests that the Mediterranean diet, particularly when enriched with antioxidant-rich supplements, may help maintain cognitive function in older age and reduce the risk of dementia. However, the effects appear to be population- and context-dependent, and further long-term randomized trials are needed to clarify the underlying mechanisms [218]. The main characteristics of the key studies are summarized in the table below (Table 2).

### 5.2. The Mediterranean Diet and the Risk of Alzheimer’s Disease: Summary of Epidemiological Evidence

Numerous prospective and cross-sectional studies have demonstrated that greater adherence to the Mediterranean diet is associated with better cognitive health and a reduced risk of developing Alzheimer’s disease. Early influential cohort studies, such as those by Gu et al. [23] and Scarmeas et al. [222], showed that higher MeDi scores were linked to a 30–50% lower risk of AD or conversion from MCI to AD, with evidence of a dose–response relationship. Similarly, Morris et al. [22] reported that higher adherence to both the Mediterranean and MIND diets was significantly associated with a lower incidence of AD.

Cross-sectional data support these findings: in the Australian study by Gardener et al. [26], individuals with AD or MCI exhibited significantly lower MeDi scores compared to healthy controls, while higher adherence was related to less decline in MMSE performance over follow-up. In very old adults, Nicoli et al. [223] found that greater adherence to the Mediterranean diet and higher consumption of plant-based foods were associated with lower prevalence and incidence of AD and dementia.

Conversely, some more recent long-term investigations—such as Glans et al. [211]—did not observe significant associations between Mediterranean diet adherence and the risk of AD or overall dementia. These inconsistencies may partly reflect methodological differences in dietary assessment and changes in lifestyle behaviors over time. Overall, current epidemiological evidence suggests that higher adherence to the Mediterranean diet—particularly greater intake of vegetables, fruits, fish, and olive oil—is associated with a significantly lower risk of Alzheimer’s disease (Table 3).

### 5.3. Mediterranean Diet and Parkinson’s Disease Risk

Epidemiological evidence increasingly indicates that higher adherence to the Mediterranean diet is associated with a lower risk of Parkinson’s disease and a reduced likelihood of developing prodromal PD features. Most cohort and case–control studies report relative risk reductions of 20–50%, with the largest benefits generally observed in European and Mediterranean populations. In Asian populations, risk reductions were somewhat smaller, although the direction of the effect—reduced Parkinson’s disease risk—remained consistent. Overall, the strongest associations were observed in Mediterranean, Caucasian, and Latin American cohorts, whereas effects in Asian and African American populations tended to be smaller or less consistent, potentially reflecting differences in baseline dietary patterns, cultural eating habits, and gut microbiome profiles.

In a U.S. cohort of 706 participants, Agarwal et al. [231] found that greater adherence to the MeDi, alongside the MIND diet, was associated with a lower risk of developing parkinsonism (HR = 0.89; 95% CI 0.83–0.96). In a case–control study of 455 individuals, Alcalay et al. [232] reported that high MeDi adherence was linked to a 14% reduction in PD risk (OR = 0.86; 95% CI 0.77–0.97; *p* = 0.01) and delayed disease onset.

Findings from the Greek HELIAD cohort further support a neuroprotective role of the MeDi. Each one-point increase in MeDi score was associated with a 2% reduction in prodromal PD probability (*p* < 0.001), and participants in the highest adherence quartile had a ~21% lower risk than those in the lowest quartile [233,234]. Longitudinal analyses showed that higher MeDi adherence reduced the likelihood of developing possible or probable prodromal PD by 60–70% (*p*-trend = 0.003) and lowered the incidence of PD/dementia with Lewy bodies by approximately 9–10% (HR = 0.906; 95% CI 0.823–0.997).

Consistent findings have been observed in Scandinavian populations. In a study of over 47,000 Swedish women, Yin et al. [235] reported that the highest MeDi adherence was associated with a 46% lower risk of PD (HR = 0.54; 95% CI 0.30–0.98), while each unit increase in MeDi score conferred a 29% lower PD probability among women aged ≥65 years (95% CI 0.57–0.89).

Large U.S. datasets provide further support. Molsberry et al. [236] examined 47,679 participants and found that individuals in the highest aMED quintile had 18–33% lower odds of exhibiting three or more prodromal PD features (OR = 0.82–0.67; *p*-trend < 0.001). Similarly, Xu et al. [237], using NHANES data, observed that high MeDi adherence was associated with a 22% reduction in PD odds (OR = 0.78; 95% CI 0.65–0.93), whereas adherence to a Western dietary pattern more than doubled PD risk (OR = 2.19; 95% CI 1.16–4.14).

Taken together, these findings consistently support an inverse association between Mediterranean diet adherence and Parkinson’s disease risk, as summarized in Table 4.

## 6. Cognitive Effects of Trans-Resveratrol

Trans-resveratrol has been investigated for its effects on cerebral perfusion, neurovascular function, and cognition in both acute and chronic settings. Acute supplementation studies [242,243,244] consistently demonstrate increased cerebral blood flow and enhanced oxygen extraction in the frontal cortex, accompanied by improved neurovascular coupling. These physiological changes were observed in healthy older adults and individuals with type 2 diabetes. Although acute administration generally did not yield significant improvements on short-term cognitive tests, it reliably supported cerebrovascular responsiveness during cognitive tasks.

Chronic supplementation trials provide stronger evidence for cognitive benefits. Longer-term administration of trans-resveratrol [157,245,246,247,248,249] has been shown to improve memory performance, particularly in domains related to retention and consolidation, while enhancing hippocampal functional connectivity. Several studies also documented increased cerebrovascular responsiveness to cognitive and hypercapnic stimulation, alongside improvements in metabolic parameters such as HbA1c and insulin resistance—factors relevant to neurodegenerative risk. Importantly, most human trials used resveratrol doses in the range of 150–1000 mg/day, which are several hundred-fold higher than the amounts obtainable from a Mediterranean dietary pattern, and therefore the clinical relevance of these findings to dietary intake is limited.

In individuals with mild cognitive impairment or Alzheimer’s disease [189,250,251,252,253,254], trans-resveratrol reduced markers of neuroinflammation, activated SIRT1-mediated neuroprotective signaling, and partially preserved hippocampal volume or structure. Cognitive gains in these populations were generally modest, but biological and neuroimaging markers suggest a slowing of neuropathological processes.

Natural dietary sources of resveratrol have also been examined. Grape-based formulations and resveratrol-enriched wine improved attention, working memory, and regional brain metabolism in both younger and older adults [255], supporting the concept that whole-food sources may exert synergistic effects.

Overall, available evidence indicates that trans-resveratrol enhances cerebrovascular function and supports neuroprotective pathways, with the most consistent benefits emerging from chronic supplementation in older adults and individuals at increased risk of cognitive decline. It is important to note that the doses of trans-resveratrol used in clinical supplementation trials (ranging from ~150 mg/day to 1000 mg/day) are substantially higher than the amounts typically obtained from a Mediterranean diet, where average daily intake through natural sources such as red grapes, red wine, and berries is estimated at approximately 1–5 mg/day. Therefore, while supplementation studies provide mechanistic and proof-of-concept evidence for cognitive and cerebrovascular benefits, these effects may not be fully achievable through habitual dietary intake alone. Future research should aim to clarify whether long-term adherence to a polyphenol-rich Mediterranean diet can produce comparable outcomes at physiologically attainable resveratrol levels. Key findings from these studies are summarized in Table 5.

## 7. Olive Oil-Derived Polyphenols and Their Role in Cognitive Health

### 7.1. Cognitive Outcomes in MCI and Mild AD Populations Following Olive Oil or Olive Extract Interventions

Several clinical trials have investigated the effects of high-polyphenol extra virgin olive oil and other olive-derived extracts on cognitive performance in adults with MCI and mild AD. Overall, regular consumption of high-polyphenol EVOO or olive extracts has been associated with improvements in cognitive function in these populations. Tsolaki et al. [197] reported that in 50 MCI participants, 12 months of high-polyphenol early harvest EVOO combined with a Mediterranean diet significantly improved MMSE, ADAS-Cog, Digit Span, and Letter Fluency scores compared to moderate-polyphenol EVOO and Mediterranean diet alone (*p* < 0.05). Dimitriadis et al. [258] found in 43 MCI participants that high-polyphenol EVOO reduced EEG-measured over-excitation, decreased the theta/beta ratio, and enhanced integrated dynamic functional connectivity (*p* < 0.001).

Kaddoumi et al. [259] conducted a 6-month study with 25 MCI participants and observed that EVOO reduced blood–brain barrier permeability, increased both resting-state and task-based functional connectivity, and improved CDR and behavioral scores. Refined olive oil (ROO) improved CDR scores and task-based activation but did not affect BBB permeability or functional connectivity. Both EVOO and ROO lowered plasma Aβ42/Aβ40 and p-tau/t-tau ratios. Since refined olive oil contains only negligible amounts of oleuropein and other polyphenols, the observed cognitive effects cannot be attributed to oleuropein, and alternative mechanisms—such as the high monounsaturated fatty acid (MUFA) content—are more likely to account for these findings. This distinction should be considered when interpreting the results.

In mild AD populations, olive leaf extract (OLE) and oleuropein + S-acetyl-glutathione supplementation preserved or improved cognitive and functional scores over 6 months [260,261]. Additionally, low-dose EVOO integrated into the Mediterranean diet improved ADAS-Cog scores over 12 months [262]. Collectively, these findings support the cognition-enhancing effects of high-polyphenol EVOO and olive-derived extracts in MCI and mild AD populations (Table 6).

### 7.2. Olive Oil Polyphenols and Cognitive Function: Evidence from Mediterranean Diet Studies

Among epidemiological and intervention studies examining the effects of the Mediterranean diet on cognitive function, several highlight the role of extra virgin olive oil and olive polyphenols, including oleuropein, in improving memory performance and global cognitive function. In the PREDIMED trial, participants consuming EVOO showed significant improvements in the MMSE and Clock Drawing Test (CDT) compared with the control group [196]. Similarly, Valls-Pedret et al. [218,263] reported that a polyphenol-rich Mediterranean diet, including olive oil, was positively associated with enhanced verbal memory, working memory, and frontal cognitive components.

Cross-sectional studies, such as Anastasiou et al. [219] and Andreu-Reinón et al. [264], have shown that higher Mediterranean diet adherence—particularly with olive oil consumption—is associated with reduced dementia risk and better memory performance. Further evidence from Bajerska et al. [265] and Talhaoui et al. [266] indicates that specific olive oil intake, independent of total diet scores, supports cognitive domains such as global cognition, visual memory, and executive function, highlighting the neuroprotective potential of olive oil and its polyphenols, including oleuropein, in older adults.

It is important to note that refined olive oil contains negligible amounts of oleuropein and other polyphenols; thus, any cognitive improvements observed with ROO are likely mediated by other components, such as monounsaturated fatty acids or minor bioactive compounds, rather than the polyphenols responsible for the effects of high-polyphenol extra virgin olive oil. While the Mediterranean diet naturally provides EVOO and polyphenols, additional supplementation can deliver higher doses of bioactive compounds, potentially enhancing neuroprotective effects, improving bioavailability, or more effectively targeting cognitive pathways.

Overall, current evidence suggests that incorporating olive oil and polyphenol-rich foods into the Mediterranean diet may be critical for preventing Alzheimer’s disease and other cognitive impairments, particularly in aging populations (Table 7).

## 8. Non-Olive Polyphenols and Cognitive Function: Evidence from Flavonoids, Catechins, and Cocoa Flavanols

This review does not provide an exhaustive analysis of trials focusing on general polyphenol supplementation or broader polyphenol subclasses (e.g., *Ginkgo biloba*, soy isoflavones, anthocyanins, cocoa flavanols, flavonoid extracts, chlorogenic acids, curcuminoids). These areas encompass more than one hundred clinical trials and have been comprehensively reviewed elsewhere. Within the scope of this article, these compounds are briefly summarized to contextualize the wider evidence base linking polyphenol intake to cognitive aging.

Multiple prospective cohort studies suggest that higher dietary intake of flavonoids and other polyphenols is associated with slower age-related cognitive decline. In the SU.VI.MAX cohort, Kesse-Guyot et al. [271] reported that higher intakes of total polyphenols—particularly catechins, theaflavins, and flavonols—were associated with better verbal memory and language performance in middle-aged adults. Similarly, in the Nurses’ Health Study, greater midlife flavonoid consumption was linked to a higher likelihood of healthy aging, including preserved cognitive and mental function [272].

These findings are consistent with results from the Framingham Offspring Study, where Shishtar et al. [273] observed a trend toward slower cognitive decline with higher flavanol intake. Data from the Memory and Aging Project further showed that dietary patterns rich in green leafy vegetables and their bioactive components [e.g., lutein, folate, vitamin K] were associated with significantly slower cognitive decline [274]. Collectively, this evidence supports the hypothesis that flavonoid- and polyphenol-rich diets during midlife may confer protection against late-life cognitive deterioration through antioxidant, anti-inflammatory, and neuroprotective mechanisms.

Human observational and interventional studies consistently indicate that flavonoids found in berries and grapes—particularly anthocyanins and flavanols—are associated with beneficial cognitive effects. In the Nurses’ Health Study, Devore et al. [275] reported that higher blueberry and strawberry consumption was linked to slower cognitive aging in older women, corresponding to approximately 2–2.5 years of cognitive “youthfulness.”

Interventional evidence complements these findings. Krikorian et al. [276] showed that 16 weeks of Concord grape juice supplementation improved memory and increased neural activation in older adults with MCI. In a six-month placebo-controlled trial, Lee et al. [254] found that freeze-dried grape powder (FDGP), rather than isolated grape polyphenols, preserved metabolic integrity in brain regions vulnerable in early Alzheimer’s disease. Additionally, long-term data from the Framingham Offspring Study revealed that higher intakes of flavonols, anthocyanins, and flavonoid polymers were associated with significantly lower risk of Alzheimer’s disease and related dementias (HR 0.24–0.58) [277]. Together, these results suggest that regular consumption of flavonoid-rich berries and grapes may slow cognitive decline and reduce neurodegenerative risk.

Catechins—major polyphenols in green tea—exhibit well-documented antioxidant, anti-inflammatory, and neuroprotective properties [278,279,280]. Clinical trials support cognitive benefits in midlife and older adulthood. In a double-blind, randomized, placebo-controlled trial, Baba et al. [281] found that 12 weeks of daily supplementation with decaffeinated green tea catechins (336 mg) improved cognitive performance in Japanese adults aged 50–69 years. Participants exhibited reduced error rates after a single dose and faster response times on working memory tasks after 12 weeks.

Consistent results were reported by Ide et al. [282], who observed improvements in overall cognitive performance—particularly attention and memory—in elderly Japanese individuals with MCI. These findings suggest that habitual intake of green tea catechins may confer modest but measurable cognitive benefits, likely mediated by antioxidant and neuroprotective mechanisms.

A growing body of evidence also supports the cognitive benefits of cocoa-derived flavanols, particularly in older adults. In the Cocoa, Cognition and Aging (CoCoA) studies, daily consumption of cocoa flavanols for eight weeks improved attention, executive function, and verbal fluency in older individuals with normal cognition and in those with MCI [283,284]. These improvements were accompanied by reductions in insulin resistance, blood pressure, and oxidative stress, suggesting that vascular and metabolic pathways may mediate the observed cognitive effects.

A retrospective clinical study in MCI patients found that higher cocoa polyphenol intake was associated with slower progression of cognitive decline [285]. Long-term cohort data also show that regular chocolate consumption is inversely correlated with cognitive decline risk, independently of age, education, and cardiovascular factors [286]. Together, these findings indicate that habitual cocoa flavanol consumption may help preserve cognitive function and support healthy brain aging.

## 9. Discussion and Conclusions

This review integrates epidemiological, clinical, and mechanistic evidence linking adherence to the Mediterranean diet with healthier cognitive aging and a reduced risk of neurodegenerative diseases. Prospective cohort studies consistently show that individuals who closely follow the Mediterranean diet experience slower cognitive decline and lower incidence of Alzheimer’s disease, Parkinson’s disease, and related dementias. Although population-specific variability exists, the overall evidence supports a protective association. Clinical interventions, particularly those incorporating high-polyphenol extra virgin olive oil, have demonstrated improvements in memory, executive function, cerebrovascular responsiveness, and disease-relevant biomarkers.

Mechanistic data provide a coherent biological rationale for these observations. Mediterranean diet components modulate multiple pathways central to neurodegenerative processes, including oxidative stress, chronic inflammation, mitochondrial dysfunction, cerebrovascular impairment, and gut–brain axis regulation. Polyphenols—especially resveratrol, oleuropein, and hydroxytyrosol—appear particularly relevant due to their roles in modulating amyloid and tau pathology, activating SIRT1, supporting mitochondrial and endothelial integrity, and regulating neuroinflammatory signaling.

It should be noted that much of the mechanistic evidence derives from in vitro studies, animal models, or metabolic biomarkers, whereas human, long-term neural outcomes (cognition, neurodegeneration, protein aggregation) remain limited. Daily intake of resveratrol and oleuropein from a typical Mediterranean diet is extremely low (~1 mg/day), suggesting that these compounds alone are unlikely to account for the observed cognitive benefits. Therefore, mechanistic and intervention studies must clearly specify the model system (cellular, animal, or human) and compare administered doses with achievable dietary intake levels. Moreover, extra virgin olive oil contains a complex mixture of polyphenols, and cognitive benefits likely reflect synergistic effects between polyphenols and other bioactive components, such as monounsaturated fatty acids. Overall, the cognitive benefits of the Mediterranean diet are probably not attributable to a single polyphenol, but rather to the combined action of multiple dietary components.

The clinical relevance of the Mediterranean diet extends beyond prevention. Its multidimensional biological effects suggest potential as a complementary therapeutic strategy for individuals with mild cognitive impairment or early-stage neurodegenerative disease. Polyphenols may also serve as biomarkers of dietary exposure and biological effect, and targeted supplementation could be considered for individuals with insufficient habitual intake. Responses to dietary and polyphenol-based interventions are likely individual-specific, influenced by genetics, comorbidities, physical activity, smoking, sleep patterns, and regional dietary habits. Collectively, the evidence supports the Mediterranean diet as a biologically plausible, feasible approach to promote cognitive health, with mechanistic and clinical data highlighting the importance of the synergistic action of multiple dietary components rather than any single polyphenol.

Interpretation of the literature must also take into account several limitations. Considerable variability exists in the scoring systems used to assess Mediterranean diet adherence, as well as in the dietary assessment tools themselves, which complicates the comparison of results across studies. Many cohort studies rely on self-reported dietary intake, which introduces recall bias. Interindividual differences in polyphenol absorption, metabolism, and bioavailability introduce further heterogeneity, as do differences in gut microbiome composition and lifestyle habits. Randomized controlled trials remain relatively few, often involve modest sample sizes and short intervention periods, and employ diverse formulations of polyphenol-rich foods or supplements. Importantly, the bioavailability of resveratrol in dietary sources is low and highly variable, and intervention doses used in mechanistic studies often exceed what can be achieved through habitual dietary intake, highlighting the need for careful interpretation of translatability. Furthermore, the studies summarized in the tables exhibit considerable heterogeneity, not only in dietary assessment methods but also due to geographical and cultural differences, which complicates direct comparison and synthesis of results. These limitations underscore the need for additional mechanistic and translational research in humans.

Future studies should aim to include larger, multicenter clinical trials with harmonized definitions of MedDiet adherence and standardized intervention protocols. The incorporation of biomarker-based endpoints—such as measures of amyloid and tau pathology, neuroinflammation, vascular function, and microbiome composition—will be crucial to clarifying causal pathways and strengthening the clinical relevance of findings. Integrating neuroimaging, metabolomics, and vascular assessments will help elucidate how specific components of the MedDiet contribute to neuroprotection and which individuals are most likely to benefit from dietary modification or targeted polyphenol supplementation. Such work is essential to advance the development of personalized nutritional strategies for neurodegenerative disease prevention and management.

In conclusion, although important gaps in our understanding remain, the collective evidence suggests that the Mediterranean diet is a promising, feasible, and biologically plausible approach to support cognitive health and reduce the burden of neurodegenerative diseases. Polyphenols—particularly those derived from extra virgin olive oil and grapes—likely contribute to these benefits through their combined antioxidant, anti-inflammatory, mitochondrial, and anti-amyloid actions, but the overall protective effect of the MedDiet is likely due to the synergistic interaction of multiple dietary components rather than the effect of any single polyphenol. Continued research focusing on biomarker-based outcomes, dose–response relationships, and precision nutrition strategies will be essential to translate these insights into targeted dietary recommendations and clinically meaningful interventions for individuals at risk of cognitive decline and neurodegeneration.

## Figures and Tables

**Figure 1 nutrients-17-03929-f001:**
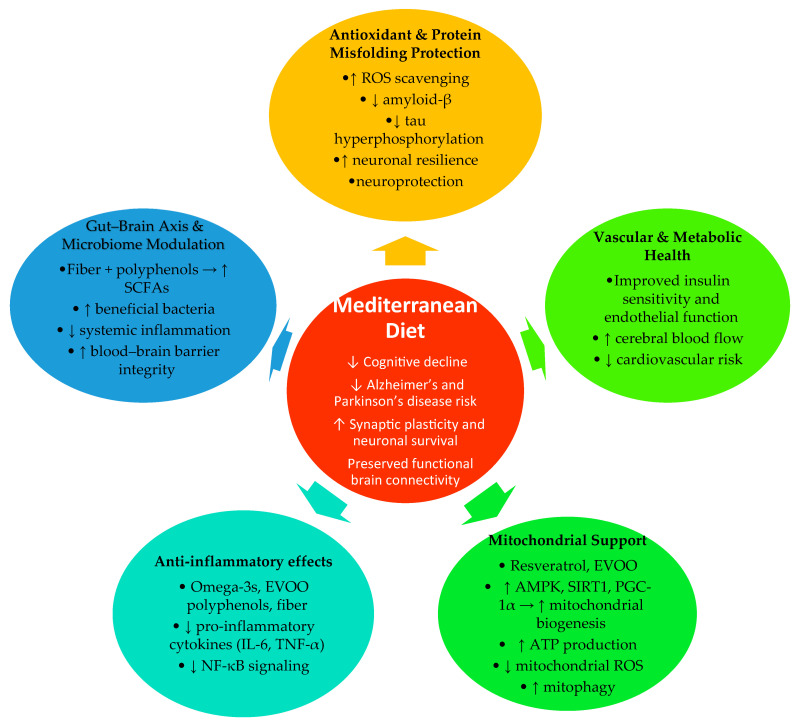
Mechanistic overview: how the Mediterranean diet may slow neurodegeneration. Abbreviations EVOO—Extra Virgin Olive Oil; IL-6—Interleukin-6; TNF-α—Tumor Necrosis Factor-alpha; NF-κB—Nuclear Factor kappa-light-chain-enhancer of activated B cells; ROS—Reactive Oxygen Species; AMPK—AMP-activated Protein Kinase; SIRT1—Sirtuin-1; PGC-1α—Peroxisome Proliferator-Activated Receptor Gamma Coactivator-1 Alpha; SCFAs—Short-Chain Fatty Acids; ↑ indicates an increase in the corresponding variable; ↓ indicates a decrease in the corresponding variable.

**Table 1 nutrients-17-03929-t001:** Comparative overview of Mediterranean, DASH, MIND, and Western dietary patterns.

Diet	Main Food Groups	Key Nutrients/Bioactive Compounds	Health Effects/Targets	Notes/Emphasis
Mediterranean (MedDiet)	Fruits, vegetables, legumes, whole grains, nuts, extra virgin olive oil (EVOO), moderate fish and poultry, small amounts of red wine	Monounsaturated fatty acids (MUFA), omega-3 fatty acids, polyphenols (resveratrol, oleuropein, oleocanthal), antioxidants, vitamins (C, E), minerals (Mg, K)	Cardiovascular health, reduced oxidative stress, neuroprotection, cognitive function support	Plant-based emphasis, EVOO as main fat source, regular fish intake, moderate meat consumption, heart-healthy fatty acid profile
DASH (Dietary Approaches to Stop Hypertension)	Vegetables, fruits, whole grains, low-fat dairy, fish, poultry, nuts, legumes	Potassium, calcium, magnesium, fiber, low saturated fat, moderate protein	Blood pressure reduction, cardiovascular risk reduction, metabolic health improvement	Low sodium, limited added sugar and processed foods, nutrient-dense, balanced macro- and micronutrients
MIND (Mediterranean-DASH Intervention for Neurodegenerative Delay)	Fruits (especially berries), vegetables (especially leafy greens), whole grains, nuts, legumes, olive oil, fish, moderate poultry, small amounts of red wine	MUFA, omega-3, polyphenols, antioxidants, vitamins (folate, B6, B12, C, E)	Neuroprotection, reduced risk of Alzheimer’s disease, slower cognitive decline	Combines features of MedDiet and DASH, emphasis on berries and leafy greens, limited red meat, butter, sweets
Western diet	Processed meats, red meat, refined grains, sugary foods and beverages, high-fat dairy, fried foods	High saturated fat, trans fat, added sugar, low fiber, vitamin and mineral deficiencies	Increased cardiometabolic risk, insulin resistance, obesity, inflammation	High-calorie, low plant-based nutrient intake, low antioxidant and micronutrient content, risk factor for chronic diseases

Abbreviations: MedDiet, Mediterranean Diet; DASH, Dietary Approaches to Stop Hypertension; MIND, Mediterranean-DASH Intervention for Neurodegenerative Delay; WESTERN, Western Dietary Pattern; EVOO, Extra Virgin Olive Oil; MUFA, Monounsaturated Fatty Acids; ROS, Reactive Oxygen Species; SIRT1, Sirtuin-1; PGC-1α, Peroxisome Proliferator-Activated Receptor Gamma Coactivator-1 Alpha.

**Table 2 nutrients-17-03929-t002:** Mediterranean dietary patterns and cognitive decline or mild cognitive impairment.

Study (Author, Year)	Design	N	Population	Exposure/Intervention	Outcome(s)	Key Findings
Haring et al., 2016 [210]	Prospective cohort, 9 y	6425	Postmenopausal women aged 65–79 y	Dietary patterns (aMED, HEI, DASH)	Cognitive decline, MCI	No significant association with MedDiet.
Hosking et al., 2019 [213]	Prospective cohort, 12 y	1220	Older adults (age ~ 62)	MIND vs. MedDiet adherence	Cognitive impairment (MCI, AD/VaD)	MIND diet associated with 19% lower odds of MCI/dementia (53% reduction in highest tertile).
Olsson et al., 2015 [220]	Prospective cohort, 12 y	1038	70-year-old men	Mediterranean-like diet	Dementia, cognitive impairment	No significant association; possible protective trend for CI (OR = 0.32).
Tsivgoulis et al., 2013 [212]	Prospective cohort, 4 y	17,478	Adults without CI; 45–98 y	MedDiet adherence (0–9)	Cognitive impairment	Higher adherence reduced CI risk (OR = 0.87); effect stronger in non-diabetics.
Trichopoulou et al., 2015 [208]	Prospective cohort, 7–17 y	401	Elderly men (n = 144) and women (n = 257), mean age 74 y	MedDiet adherence (MDS 0–9)	Cognitive decline (MMSE)	Higher MDS → lower cognitive decline (MDS 6–9: OR 0.46 mild, 0.34 substantial); strongest in ≥75 y; vegetables and healthy fats most protective.
Bhushan et al., 2018 [209]	Prospective cohort, long-term	27,842	Adult men (mean age 64.4 y)	MedDiet score	Subjective cognitive function	Highest adherence linked to lower odds of poor SCF (OR = 0.64).
Tor-Roca et al., 2023 [214]	Nested case–control, 12 y	840	Older adults free of dementia	Metabolomic MedDiet score	Cognitive decline	Higher MDMS linked to lower odds of decline (OR = 0.90).
Feng et al., 2024 [215]	Prospective cohort, 3 y	3961	Rural elderly ≥ 65 y	MedDiet adherence (MEDAS)	Cognitive decline (MMSE drop ≥2)	High adherence reduced decline (β = −0.020), significant in women; beans, fish, cooked vegetables protective.
Allcock et al., 2022 [217]	Cross-sectional	294	Age 70.4 ± 6.2, 68% female	MEDAS	Cognitive risk (AD8), functional ability (iADL)	Higher adherence improved function (β = 0.172) and reduced cognitive risk (β = −0.134); not significant in cognitively intact adults.
Chan et al., 2013 [216]	Cross-sectional	3670	Older Chinese adults ≥ 65 y	Dietary patterns (MDS, factor analysis)	Cognitive impairment (CSI-D)	In women, “vegetables-fruits” (OR 0.73) and “snacks-drinks-milk” patterns (OR 0.65) reduced CI risk; no effect in men.
Godos et al., 2023 [31]	Cross-sectional	883	Middle-aged and older adults (55–75 y)	MedDiet adherence (FFQ quartiles)	Cognitive status, mental health, QoL, successful aging	Highest adherence reduced cognitive impairment (OR = 0.19), depressive symptoms (OR = 0.19), improved QoL (OR = 14.04) and successful aging (OR = 1.65).
Martínez-Lapiscina et al., 2013 [196]	RCT, 6.5 y	522 (cognitive testing)/1055 total	Older adults at high cardiovascular risk, mean age 74.6, 44.6% men	MedDiet + EVOO or Nuts vs. low-fat diet	Global cognition (MMSE, Clock Drawing)	Both interventions improved MMSE and CDT scores compared with control.
Roberts et al., 2010 [221]	Cross-sectional	1233	U.S. adults aged 70–89 y	Mediterranean diet components	Mild cognitive impairment	High vegetable intake (OR = 0.66) and favorable unsaturated/saturated fat ratio (OR = 0.52) associated with lower MCI risk.

Abbreviations: AD—Alzheimer’s disease; aMED—Alternative Mediterranean Diet; CDT—Clock Drawing Test; CI—Cognitive impairment; DASH—Dietary Approaches to Stop Hypertension; FFQ—Food Frequency Questionnaire; HEI—Healthy Eating Index; iADL—Instrumental Activities of Daily Living; LCHP—Low-carbohydrate high-protein diet; MCI—Mild cognitive impairment; MDS—Mediterranean Diet Score; MEDAS—Mediterranean Diet Adherence Screener; mMDS—Modified Mediterranean Diet Score; MMSE—Mini-Mental State Examination; QoL—Quality of Life; RCT—Randomized Controlled Trial; VaD—Vascular dementia; WHO—World Health Organization.

**Table 3 nutrients-17-03929-t003:** Summary of epidemiological studies on Mediterranean and related dietary patterns in relation to Alzheimer’s disease and dementia risk.

Study (Author, Year)	Design	N	Population	Exposure/Intervention	Outcome(s)	Key Findings
de Crom et al., 2022 [224]	Prospective cohort (Rotterdam Study)	5375/2861	Adults ≥ 55 y, Netherlands	MIND diet adherence	All-cause dementia	Higher MIND adherence associated with lower dementia risk (HR 0.76–0.85); associations attenuated over time.
Gu et al., 2010 [23]	Prospective cohort	1219	Older adults ≥ 65 y, non-demented	Mediterranean diet (MeDi score)	Alzheimer’s disease	Higher MeDi adherence reduced AD risk by 34%; associated with lower hsCRP, not mediated by biomarkers.
Morris et al., 2015 [22]	Prospective cohort	923	Adults 58–98 y	MIND, Mediterranean, DASH diets	Alzheimer’s disease	Higher MIND adherence lowered AD risk (HR 0.47–0.65); MeDi and DASH protective only in highest tertiles.
Scarmeas et al., 2009 [222]	Prospective cohort	1875 (1393 cognitively normal; 482 MCI)	Multiethnic older adults; 76.9 (6.5) y	Mediterranean diet adherence (0–9 scale)	Incidence of MCI and progression MCI → AD	Higher MeDi adherence reduced MCI risk (HR 0.72) and MCI → AD conversion (HR 0.52); dose–response trend observed.
Gardener et al., 2012 [26]	Cross-sectional (AIBL Study)	970 (723 HC, 98 MCI, 149 AD)	Older adults: 149 AD, 98 MCI, 723 healthy controls	Mediterranean diet adherence (0–9 scale, FFQ-based)	AD and MCI status, MMSE change	Lower MeDi adherence in AD and MCI vs. controls (*p* < 0.001, *p* < 0.05). Higher adherence linked to less MMSE decline over 18 months.
Nicoli et al., 2021 [223]	Population-based cross-sectional and longitudinal study (Monzino 80-plus)	1390 (cross-sectional); 512 (longitudinal)	Adults ≥ 80 y, Varese province, Italy	Mediterranean diet adherence and components (FFQ-based tertiles)	Prevalent and incident dementia	Higher MeDi adherence and greater intake of fruits, vegetables, legumes, and total food associated with lower dementia prevalence and incidence (HR ≈ 0.66–0.68). Reverse causality possible.
Glans et al., 2023 [211]	Prospective cohort (Malmö Diet and Cancer Study)	28,025	Adults born 1923–1950, Sweden	Conventional dietary recommendations; modified Mediterranean diet	All-cause dementia, AD, VaD, CSF Aβ42	Neither adherence to conventional diet nor modified Mediterranean diet was associated with lower risk of all-cause dementia, AD, VaD, or Aβ accumulation over ~20 years follow-up.
Mamalaki et al., 2022 [225]	Prospective cohort (HELIAD Study)	1018	Adults ≥ 65 y, Greece	Total Lifestyle Index (Mediterranean diet, physical activity, sleep, daily activities)	Cognitive decline and dementia	Higher TLI associated with slower cognitive decline and lower dementia risk (0.5% less decline/year per unit; 0.2% reduced dementia risk/year per unit); sleep was the exception.
Shannon et al., 2023 [226]	Prospective cohort (UK Biobank)	60,298	Dementia-free adults at baseline (mean age 63.8 y), followed for mean 9.1 years	Mediterranean diet adherence (MEDAS and PYRAMID scores)	Incident all-cause dementia	Higher MeDi adherence associated with lower dementia risk (MEDAS HR 0.77; PYRAMID HR 0.86); effect independent of genetic risk.
Gu et al., 2010 [227]	Prospective cohort	2148	Community-dwelling elderly ≥ 65 y, New York	Dietary pattern derived from AD-related nutrients (RRR-based)	Incident Alzheimer’s disease	Highest adherence to protective dietary pattern (nuts, fish, vegetables, fruits, poultry; low red/high-fat meat, butter) associated with lower AD risk (HR 0.62; 95% CI 0.43–0.89).
Kheirouri et al., 2024 [228]	Case–control	89 (60 AD, 29 healthy)	Older adults with AD and age-matched healthy controls	MIND and Mediterranean (MeDi) dietary patterns	Alzheimer’s disease	Higher MIND adherence associated with 40% lower AD risk per unit increase; MeDi pattern showed a non-significant 14% risk reduction; MIND and MeDi scores not strongly correlated with MMSE scores.
Charisis et al., 2021 [229]	Prospective cohort (HELIAD)	391	Non-demented older adults ≥ 64 y, Greece	Baseline plasma glutathione (GSH)	Incident Alzheimer’s disease, cognitive decline	Highest GSH tertile had 70% lower AD risk vs. lowest; slower executive function decline over ~3 years; dose–response trend observed.
Calil et al., 2018 [230]	Cross-sectional	96	Elderly ≥60 years (NC, MCI, AD), neurology outpatient clinic, Brazil	Mediterranean and MIND diet adherence	Cognitive performance (MMSE, BCSB)	Higher adherence linked to better cognition only in healthy controls; no effect in MCI or AD.

Abbreviations: AD—Alzheimer’s disease; Aβ—Beta-amyloid; AIBL—Australian Imaging, Biomarkers and Lifestyle Study; BCSB—Brief Cognitive Screening Battery; CI—Cognitive impairment; CSF—Cerebrospinal fluid; DASH—Dietary Approaches to Stop Hypertension; FFQ—Food Frequency Questionnaire; GSH—Glutathione; HC—Healthy controls; HR—Hazard ratio; hsCRP—High-sensitivity C-reactive protein; MCI—Mild cognitive impairment; MeDi—Mediterranean diet; MEDAS—Mediterranean Diet Adherence Screener; MMSE—Mini-Mental State Examination; MIND—Mediterranean-DASH Intervention for Neurodegenerative Delay; NC—Normal cognition; PYRAMID—Mediterranean diet adherence score [UK Biobank]; RRR—Reduced Rank Regression; TLI—Total Lifestyle Index; VaD—Vascular dementia.

**Table 4 nutrients-17-03929-t004:** Summary of observational studies investigating adherence to Mediterranean diet and Parkinson’s disease risk or prodromal features.

Study (Author, Year)	Design	N	Population	Exposure/Intervention	Outcome(s)	Key Findings
Agarwal et al., 2018 [231]	Prospective cohort	706	Older adults (59–97 y), free of parkinsonism	MIND, Mediterranean, DASH	Incident parkinsonism	MIND diet inversely associated with parkinsonism (HR = 0.89; 95% CI 0.83–0.96); Mediterranean diet marginally protective; DASH not associated.
Alcalay et al., 2012 [232]	Case–control	455	PD: 68.2 ± 11.0 yrs; Controls: 72.4 ± 9.6 yrs	Mediterranean diet (9-point scale)	PD status, age at onset	Higher adherence linked to lower PD odds (OR = 0.86; 95% CI 0.77–0.97; *p* = 0.01) and later onset.
Keramati et al., 2024 [238]	Cross-sectional	170	Patients: 60.8 ± 9.8 yrs; Controls: 60.4 ± 9.8 yrs	DASH, Mediterranean, MIND	PD risk and severity (UPDRS)	DASH inversely associated with PD risk (OR = 0.86; 95% CI 0.75–0.98); Mediterranean and MIND not significant.
Maraki MI et al., 2019 [233]	Population cohort	1731	Older adults (≥65 y, Greece)	Mediterranean diet (0–55 score)	Prodromal PD probability	Higher adherence reduced pPD probability (*p* < 0.001); 2% lower risk per unit increase; top quartile ~21% lower probability.
Maraki MI et al., 2023 [234]	Longitudinal cohort	1047	Older adults (≥65 y, Greece)	Mediterranean diet (0–55 score)	Prodromal PD/PD–DLB incidence	Higher adherence reduced pPD progression (β = −0.003; *p* = 0.010) and PD/DLB risk (HR = 0.91; 95% CI 0.82–1.00; *p* = 0.044).
Molsberry S et al., 2020 [236]	Prospective cohort	17,400 (completed secondary screening)	Middle-aged and older adults, female nurses (NHS) and male health professionals (HPFS), without PD	Alternate Mediterranean diet (aMED), AHEI	Prodromal PD features	High aMED/AHEI adherence linked to fewer prodromal features (OR = 0.82 for ≥3 vs. 0 features).
Okubo H et al., 2012 [239]	Case–control	617	Japanese adults PD Cases: 68.5 ± 8.6; Controls: 66.6 ± 8.5 y	Dietary patterns (factor analysis)	PD risk	“Healthy” pattern (vegetables, fish, fruit) inversely related to PD (OR = 0.54; 95% CI 0.32–0.92).
Strikwerda AJ et al., 2021 [240]	Prospective cohort	9414	Dutch adults, PD-free at baseline; Median 62.2 y (IQR 58–70)	Mediterranean diet, Dutch diet quality	Incident PD	Mediterranean pattern suggested lower PD risk (HR = 0.89; 95% CI 0.74–1.07), though not significant.
Xu S et al., 2023 [237]	Cross-sectional (NHANES 2015–2018)	5824 (91 PD cases)	U.S. adults ≥ 50 y	Mediterranean and Western patterns	PD diagnosis	Mediterranean diet reduced PD odds (OR = 0.78; 95% CI 0.65–0.93); Western pattern increased odds (OR = 2.19).
Yin W et al., 2021 [235]	Prospective cohort	47,128	Swedish women; Mean 39.7 at enrollment; follow-up from age 50	Mediterranean dietary pattern (MDP)	Incident PD	High adherence inversely associated (HR = 0.54; 95% CI 0.30–0.98); each unit ↑ in MDP → 29% lower PD risk ≥ 65 y.
Zhang X et al., 2022 [241]	Cross-sectional (from ongoing prospective cohorts)	71,640	Chinese adults; Mean 50.8 ± ~14 years	aMED, mAHEI	Prodromal PD features	Higher mAHEI linked to lower odds of ≥2 prodromal features (OR = 0.64; 95% CI 0.49–0.85; *p* = 0.003); aMED marginally inverse (OR = 0.74).

Abbreviations: PD—Parkinson’s disease; pPD—prodromal Parkinson’s disease; DLB—dementia with Lewy bodies; MeDi—Mediterranean diet; aMED—alternate Mediterranean diet; MDP—Mediterranean dietary pattern; mAHEI—modified Alternative Healthy Eating Index; MIND—Mediterranean-DASH Intervention for Neurodegenerative Delay; DASH—Dietary Approaches to Stop Hypertension; AHEI—Alternative Healthy Eating Index; FFQ—food frequency questionnaire; HR—hazard ratio; OR—odds ratio; β—regression coefficient; CI—confidence interval; PCA—principal component analysis; ↑ indicates an increase in the corresponding variable.

**Table 5 nutrients-17-03929-t005:** Summary of human clinical trials investigating resveratrol and cognitive or cerebrovascular outcomes.

Study (Author, Year)	Design	N	Population	Exposure/Intervention	Outcome(s)	Key Findings
Kennedy et al., 2010 [242]	Randomized, double-blind, placebo-controlled, crossover trial	22	Healthy adult men (mean age 24.8 y)	Single oral doses of trans-resveratrol (250 mg and 500 mg) vs. placebo	Cerebral blood flow (NIRS), cognitive performance	Dose-dependent increases in cerebral blood flow and oxygen extraction in the frontal cortex; no significant cognitive effects; plasma metabolites confirmed absorption.
Wightman et al., 2014 [243]	Randomized, double-blind, placebo-controlled, crossover trial	23	Healthy adults (mean age 21 y; 4 males, 19 females)	Single doses of trans-resveratrol (250 mg) alone or with piperine (20 mg) vs. placebo	Cerebral blood flow (NIRS), cognitive performance, mood, blood pressure	Co-supplementation with piperine enhanced resveratrol-induced cerebral blood flow during cognitive tasks; no significant cognitive, mood, or blood pressure effects; similar plasma metabolite levels suggest improved bioefficacy without altered bioavailability.
Wong et al., 2016 [244]	Randomized, placebo-controlled, crossover trial	36	Adults (40–80 yrs) with type 2 diabetes mellitus	Single doses of resveratrol (0, 75, 150, 300 mg) at weekly intervals	Cerebrovascular responsiveness (CVR) to cognitive and hypercapnic stimuli; cognitive performance; plasma resveratrol levels	A single 75 mg dose significantly improved neurovascular coupling and multi-tasking performance; effects correlated with plasma resveratrol, indicating improved cerebral perfusion and acute cognitive benefit in T2DM.
Witte et al., 2014 [157]	Randomized, placebo-controlled, parallel-group trial	46	Healthy overweight older adults (50–75 yrs)	200 mg/day resveratrol for 26 weeks vs. placebo	Memory performance, hippocampal functional connectivity, glucose and lipid metabolism	Improved word retention and hippocampal connectivity, reduced HbA1c and body fat; memory and connectivity changes correlated with HbA1c improvements, suggesting enhanced glucose metabolism and neuroplasticity.
Evans et al., 2017 [245]	Randomized, double-blind, placebo-controlled intervention trial	79	Postmenopausal women (45–85 yrs)	75 mg trans-resveratrol twice daily vs. placebo	Cognitive performance, cerebrovascular responsiveness (CVR), mood	Increased CVR (+17%) to hypercapnic and cognitive stimuli; improved verbal memory and overall cognitive performance correlated with CVR enhancement.
Wightman et al., 2015 [246]	Randomized, double-blind, placebo-controlled, parallel-group trial	46	Healthy young adults (18–30 yrs)	500 mg trans-resveratrol for 28 days vs. placebo	Cognitive performance, cerebral blood flow (NIRS, TCD), mood, sleep, health	Chronic resveratrol modulated cerebral blood flow acutely (day 1) and increased diastolic BP after 28 days. Minimal cognitive effects; slight improvement in 3-Back task and reduced fatigue, suggesting mild psychophysiological benefit.
Huhn et al., 2018 [247]	Randomized, double-blind, placebo-controlled trial	60	Healthy older adults (60–79 yrs)	Resveratrol 200 mg/day for 26 weeks vs. placebo	Memory (CVLT, ModBent), hippocampal connectivity/microstructure, blood biomarkers	No significant verbal memory improvement; trend for preserved pattern recognition memory. Exploratory changes in body fat, glucose, inflammatory markers, and hippocampal microstructure.
Thaung Zaw et al., 2020 [248]	Randomized, double-blind, placebo-controlled trial	129	Postmenopausal women (45–85 yrs)	75 mg trans-resveratrol twice daily for 12 months vs. placebo	Cognitive performance, cerebral blood flow, CVR, cardiometabolic markers	Resveratrol improved overall cognitive performance (*p* < 0.001) and attenuated decline in CVR to cognitive stimuli (*p* = 0.038). Long-term supplementation shows sustained cerebrovascular and cognitive benefits.
Thaung Zaw et al., 2021 [249]	Randomized, double-blind, placebo-controlled, crossover trial	125	Postmenopausal women (45–85 yrs)	75 mg trans-resveratrol twice daily for 12 months, then crossover	Cognitive performance, CBFV, CVR, cardiometabolic markers	Improved overall cognition by 33% (d = 0.17, *p* = 0.005), greater verbal memory in ≥65 yr; secondary outcomes also improved, supporting long-term cerebrovascular/cognitive benefits.
Turner et al., 2015 [250]	Randomized, double-blind, placebo-controlled, multicenter phase 2 trial	119	Patients with mild to moderate Alzheimer’s disease	Oral resveratrol 500 mg/day, titrated up to 1000 mg twice daily for 52 weeks vs. placebo	CSF and plasma Aβ40/42, tau, phospho-tau181, MRI brain volume, safety/tolerability	Resveratrol and metabolites were detectable in plasma and CSF, indicating BBB penetration. Treatment slowed decline in CSF and plasma Aβ40 vs. placebo but increased brain volume loss.
Köbe et al., 2017 [251]	Randomized, double-blind, placebo-controlled interventional study	40	Patients with mild cognitive impairment (50–80 yrs)	Resveratrol 200 mg/day for 26 weeks vs. placebo (olive oil)	Glucose control (HbA1c), hippocampal volume, microstructure, RSFC, memory performance	Resveratrol reduced HbA1c moderately (d = 0.66), increased RSFC between right anterior hippocampus and right angular cortex (*p* < 0.001), and preserved left anterior hippocampus volume (d = 0.68).
Gu et al., 2021 [252]	Randomized, double-blind trial	30	Patients with mild to moderate Alzheimer’s disease	Trans-resveratrol 500 mg/day orally for 52 weeks vs. placebo	Plasma and CSF Aβ40/Aβ42, brain volume (MRI), CSF MMP-9	Trans-resveratrol prevented decline in CSF and plasma Aβ40 seen in placebo (*p* < 0.05), reduced CSF MMP-9 levels by 46% (*p* = 0.033), and reduced brain volume loss compared with placebo (*p* = 0.011).
Zhu et al., 2018 [253]	Randomized, double-blind, placebo-controlled pilot trial	39	Patients with mild to moderate Alzheimer’s disease	Resveratrol with glucose and malate (RGM: 5 mg resveratrol + 5 g glucose + 5 g malate) twice daily for 12 months vs. placebo	ADAS-cog, MMSE, ADCS-ADL, NPI	RGM was safe and well-tolerated. Trends toward less cognitive and functional decline compared with placebo were observed, but differences were not statistically significant. Larger trials needed to assess efficacy.
Moussa et al., 2017 [189]	Retrospective analysis of a 52-week randomized, double-blind, placebo-controlled trial	38 (subset)	Mild to moderate Alzheimer’s disease (CSF Aβ42 <600 ng/mL)	Resveratrol up to 1 g orally twice daily vs. placebo	CSF and plasma biomarkers (MMPs, cytokines), MMSE, ADL scores	Resveratrol reduced CSF MMP9, modulated neuroinflammation, increased adaptive immunity markers (MDC, IL-4, FGF-2), and attenuated declines in MMSE, ADL, and CSF Aβ42 levels. Suggests SIRT1-mediated neuroprotective effects.
Moran et al., 2018 [256]	Randomized, double-blind, controlled trial	37	Older adults (68–83 yrs) with normal cognition	Daily multi-ingredient supplement for 6 months: 3000 mg omega-3 PUFAs (DHA + EPA), 10 μg vitamin D3, 150 mg resveratrol, 8 g whey protein vs. placebo	Cognitive function (executive function, memory, attention, sensorimotor speed), Stroop Color-Word Test	Overall cognitive performance did not significantly differ from placebo; intervention improved Stroop Color-Word completion time at 3- and 6-month follow-ups, suggesting limited domain-specific benefit from multi-nutrient supplementation in healthy older adults.
Scholey et al., 2014 [255]	Double-blind, balanced, crossover trial	16	Older adults (mean 70.4 yrs)	100 mL red wine vs. same wine enriched with 200 mg resveratrol	Cognitive performance (Serial Threes, Serial Sevens, RVIP), mental fatigue, serum resveratrol	Resveratrol-enriched wine improved Serial Sevens performance; red wine alone improved Serial Threes. Serum resveratrol metabolites confirmed absorption. Suggests differential cognitive effects of resveratrol vs. alcohol; replication with inert control needed
Wong et al., 2013 [257]	Randomized, double-blind, placebo-controlled crossover trial	28	Healthy obese adults (BMI 33.3 ± 0.6 kg/m^2^)	Daily 75 mg trans-resveratrol for 6 weeks vs. placebo	Brachial artery flow-mediated dilatation (FMD), blood pressure, arterial compliance, Stroop test	Chronic resveratrol increased FMD by 23% vs. placebo (*p* = 0.021); acute dose after chronic supplementation enhanced FMD by 35%.

Abbreviations: ADAS-cog—Alzheimer’s Disease Assessment Scale Cognitive Subscale; ADCS-ADL—Alzheimer’s Disease Cooperative Study Activities of Daily Living; Aβ—Amyloid beta; BP—Blood pressure; CBFV—Cerebral blood flow velocity; CVLT—California Verbal Learning Test; CVR—Cerebrovascular responsiveness; DHA—Docosahexaenoic acid; EPA—Eicosapentaenoic acid; FDG-PET—Fluorodeoxyglucose positron emission tomography; FMD—Flow-mediated dilatation; HbA1c—Hemoglobin A1c; IL-4—Interleukin-4; MDC—Macrophage-derived chemokine; MMSE—Mini-Mental State Examination; MMP—Matrix metalloproteinase; NIRS—Near-infrared spectroscopy; NPI—Neuropsychiatric Inventory; RSFC—Resting-state functional connectivity; RGM—Resveratrol, glucose, and malate combination; TCD—Transcranial Doppler; T2DM—Type 2 diabetes mellitus; RVIP—Rapid Visual Information Processing; ModBent—Modified Benton Visual Retention Test.

**Table 6 nutrients-17-03929-t006:** Human randomized controlled trials using olive-derived interventions and cognitive outcomes.

Study (Author, Year)	Population (n)	Exposure/Intervention	Control/Comparator	Duration	Main Findings
Tsolaki et al., 2020 [197]	MCI (n = 50; 54 randomized, 50 completed)	Greek High Phenolic Early Harvest EVOO (50 mL/day) + MeDi	Moderate Phenolic EVOO 50 mL/day + MeDi; MeDi alone	12 mo	HP-EH-EVOO improved MMSE, ADAS-Cog, Digit Span, Letter Fluency vs. MP-EVOO and MeDi (*p* < 0.01–0.05); benefits observed independent of APOEε4 status.
Dimitriadis et al., 2021 [258]	MCI (n = 43; MeDi 14, MP-EVOO 16, HP-EH-EVOO 13)	Greek High Phenolic Early Harvest EVOO (HP-EH-EVOO, 50 mL/day) + MeDi	Moderate Phenolic EVOO 50 mL/day + MeDi; MeDi alone	12 mo	HP-EH-EVOO reduced EEG over-excitation (ΔNI), decreased theta/beta ratio, altered EEG power spectrum, and improved integrated dynamic functional connectivity vs. MP-EVOO and MeDi (*p* < 0.001–0.0001).
Kaddoumi et al., 2022 [259]	MCI (n = 25; EVOO 13, ROO 12)	Extra Virgin Olive Oil (EVOO, 30 mL/day, 1200 mg/kg polyphenols; rich in oleocanthal 621 mg/kg and oleacein 344 mg/kg)	Refined Olive Oil (ROO, 30 mL/day, 0 polyphenols)	6 mo	EVOO reduced BBB permeability, enhanced resting-state functional connectivity, improved task-based fMRI activation, and improved CDR and behavioral scores. ROO improved CDR and task activation but did not affect BBB or connectivity. Both EVOO and ROO lowered plasma Aβ42/Aβ40 and p-tau/t-tau ratios.
Loukou et al., 2024 [260]	Mild AD (n = 55)	Olive leaf extract beverage (oleuropein 2–4 g/100 g) + MeDi	MeDi only	6 mo	OLE prevented MMSE decline and improved ADAS-Cog and functional scores.
Marianetti et al., 2022 [261]	Mild AD (n = 18; crossover)	Oleuropein 80 mg + S-acetyl glutathione 50 mg b.i.d.	No treatment (crossover)	6 mo active/6 mo washout	Stabilized or improved cognition (MMSE +8%, FAB +28%, NPI −46%, *p* < 0.01); supports antioxidant–antiamyloid synergy.
Mazza et al., 2018 [262]	Elderly (n = 180)	MedDiet + EVOO 20–30 g/day (replacing all vegetable oils)	MedDiet only	12 mo	ADAS-Cog improved more in EVOO group (−3.0 ± 0.4 vs. −1.6 ± 0.4; *p* = 0.024); suggests short-term neuroprotective effect of low-dose EVOO.

Abbreviations: MCI—Mild Cognitive Impairment, AD—Alzheimer’s Disease, EVOO—Extra Virgin Olive Oil, HP-EH-EVOO—High Phenolic Early Harvest Extra Virgin Olive Oil, MP-EVOO—Moderate Phenolic Extra Virgin Olive Oil, ROO—Refined Olive Oil, MeDi—Mediterranean Diet, OLE—Olive Leaf Extract, MMSE—Mini-Mental State Examination, ADAS-Cog—Alzheimer’s Disease Assessment Scale Cognitive Subscale, CDR—Clinical Dementia Rating, EEG—Electroencephalography, ΔNI—Change in Nonlinearity Index, FAB—Frontal Assessment Battery, NPI—Neuropsychiatric Inventory, BBB—Blood–Brain Barrier, fMRI—Functional Magnetic Resonance Imaging, b.i.d.—Twice Daily.

**Table 7 nutrients-17-03929-t007:** Studies on Mediterranean diet, olive-derived polyphenols [including oleuropein], and cognitive function.

Study (Author, Year)	Population (n)	Exposure/Intervention	Control/Comparator	Duration	Main Findings
Martínez-Lapiscina et al., 2013 [267]	Elderly, high vascular risk (n = 522)	MedDiet + EVOO or MedDiet + Nuts	Low-fat control diet	6.5 y	MedDiet + EVOO improved MMSE (+0.62, *p* = 0.005) and CDT (+0.51, *p* = 0.001); MedDiet + Nuts also improved cognition (MMSE +0.57, CDT +0.33, *p* < 0.05) vs. control.
Valls-Pedret et al., 2012 [263]	Elderly, high cardiovascular risk (n = 447)	Mediterranean diet with polyphenol-rich foods (olive oil, nuts, wine, coffee, walnuts)	Observational comparison	Cross-sectional	Higher polyphenol intake and urinary polyphenols associated with better cognitive function (MMSE, verbal memory, working memory; *p* < 0.05).
Valls-Pedret et al., 2015 [218]	Elderly, high cardiovascular risk (n = 447)	MedDiet + EVOO (1 L/wk) or MedDiet + Nuts (30 g/d)	Low-fat control diet	Median 4.1 y	MedDiet + EVOO improved RAVLT (*p* = 0.049) and Color Trail Test 2 (*p* = 0.04); MedDiet + Nuts improved memory composite (*p* = 0.04); EVOO group improved frontal and global cognition vs. controls (*p* = 0.003–0.005).
Anastasiou et al., 2017 [219]	Greek elderly (n = 1865; mean age 73 y)	Adherence to Mediterranean diet (MedDietScore 0–55)	Lower adherence	Cross-sectional baseline	Each 1-unit MedDietScore increase linked to 10% lower odds of dementia. Higher adherence associated with better memory, language, visuospatial, and composite cognition; strongest for memory. Fish and non-refined cereals particularly beneficial.
Andreu-Reinón et al., 2021 [264]	EPIC-Spain Dementia Cohort: 16,160 adults (age 30–70)	MedDiet adherence (rMED 0–18)	Lower adherence	Mean 21.6 ± 3.4 y	High rMED adherence associated with 20% lower risk of dementia (HR = 0.80). Each 2-point rMED increment reduced risk by 8% (HR = 0.92, *p* = 0.021). Stronger effects in women (non-AD dementia) and low-education participants.
Galbete et al., 2015 [268]	823 Spanish adults (mean age 62 ± 6 y)	MedDiet adherence (Trichopoulou 0–9 score)	Lower adherence	Mean 6–8 y	Higher MedDiet adherence linked to smaller cognitive decline (TICS-m difference −0.56 points, 95% CI −0.99 to −0.13). Protective effect was small but present.
Kesse-Guyot et al., 2013 [269]	3083 French adults (mean age 52 y baseline, 65 y cognitive eval)	MedDiet adherence (MDS 0–9; MSDPS 0–100)	Lower adherence	13 y	Overall, no significant association with cognition. Small effects for phonemic fluency and backward digit span; low MDS linked to lower composite cognition only in manual workers. No interaction with education.
Fischer et al., 2018 [270]	2622 German adults aged 75+ (418 incident AD cases)	Single foods: red wine, white wine, coffee, green tea, olive oil, fresh fish, fruits/vegetables, red meat and sausages	Lower or no intake	10 y	Only higher red wine intake associated with lower AD incidence, but only in men (HR = 0.82). In women, higher red wine linked to higher AD risk (HR = 1.15) and higher white wine intake with memory decline. No other foods protective. Gender-specific effects noted; APOE ε4 considered.
Talhaoui et al., 2023 [266]	172 elderly Moroccans (56.4% men)	MedDiet adherence (7 main foods + 3 less frequent foods); Olive oil separately analyzed	Lower/no adherence	Cross-sectional	Overall MedDiet adherence not associated with lower cognitive impairment risk (ORa = 0.928, 95% CI (0.831–1.037). Only olive oil protective (ORa = 0.882, 95% CI (0.815–0.953). CI more frequent in women, low education, low FFM, high BMI/fat mass, low PA.
Bajerska et al., 2014 [265]	Polish elderly >60 y, high metabolic syndrome risk, rural	MedDiet adherence; frequency of vegetables, fish, olive/rapeseed oil	Lower adherence/lower food frequency	Cross-sectional	Higher MedDiet adherence and olive/fish/vegetable intake linked to better global cognition, visual memory, attention, and executive function.

Abbreviations: AD—Alzheimer’s disease, CDT—Clock Drawing Test, CVLT-II—California Verbal Learning Test II, EVOO—Extra Virgin Olive Oil, FFM—Fat-Free Mass, HR—Hazard Ratio, MDS—Mediterranean Diet Score, MMSE—Mini-Mental State Examination, MSDPS—Mediterranean-Style Dietary Pattern Score, PA—Physical Activity, rMED—relative Mediterranean Diet Score, TICS-m—Telephone Interview for Cognitive Status modified.

## Data Availability

No new data were created or analyzed in this study.

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
