# Peer review of "Mediterranean Diet, Polyphenols, and Neuroprotection: Mechanistic Insights into Resveratrol and Oleuropein"

_nutrients, 2025, doi:10.3390/nu17243929_

Round 1
Reviewer 1 Report
Comments and Suggestions for Authors
The narrative review presents an overview on the neuroprotective role of Mediterranean diet polyphenols, integrating clinical perspectives, epidemiological and mechanistic. The different tables summarizing observational and interventional studies are particularly useful for readers. The review is well referenced and reflects an updated understanding of neurodegenerative pathways. Despite these considerations, certain aspects of the study, require elucidation before publication:
- It is recommended to further discuss the limitations of the study, such as the question regarding the bioavailability of resveratrol.
- It is recommended to better discuss the great heterogeneity of the studies indicated in the tables, such as differences in dietary assessment methods, geographical and cultural differences.
- It is recommended to summarize the review to make it more readable, especially in overlapping or redundant parts.
- It is recommended to write a more critical discussion
- It is recommended to include summary graphs or cartoons, if possible.
Author Response
Dear Reviewer,
We sincerely thank you for your extremely valuable, thorough, and constructive feedback. We greatly appreciate the level of detail in your evaluation of our work, and we are particularly pleased that you highlighted the strengths of our review - namely, the integration of clinical, epidemiological, and mechani stic perspectives, as well as the utility of the tables.
Your suggestions were entirely well-founded, and we have incorporated them into the manuscript. We have elaborated in more detail on the limitations of the narrative review, with particular attention to the bioavailability of resveratrol. The heterogeneity of the studies presented in the tables - including differences in dietary assessment methods as well as geographical and cultural variability - has also been more clearly addressed in the limitations section. Furthermore, we have condensed the text to remove redundancies and aimed to present a more critical and balanced discussion. A summary figure and table have also been added to the manuscript.
Once again, we are sincerely grateful for your careful work and forward-looking suggestions, which have greatly contributed to improving the quality of our manuscript.
With kind regards and respect,
The Authors
Reviewer 2 Report
Comments and Suggestions for Authors
Overall, this is a well-written, well-referenced and useful review detailing the results of cross-sectional, prospective, case control and RCT studies looking at the effects of a Mediterranean diet (MedDiet) on cognitive function and related parameters in the context of aging and age-related neurodegenerative disease. I found the Tables to be especially helpful. However, they should all have the same sets of information and there are overlaps between several of the Tables where the same studies are cited multiple times that should be corrected. My major concern with the review is the focus on resveratrol and oleuropein. At least with respect to resveratrol, the levels are very low in grapes, red wine and berries. For example, in grapes, resveratrol represents only ~0.2% of the total polyphenol content and red wine has ~1.9 mg/liter. Moreover, it is estimated that a person eating a MedDiet would consume ~1 mg resveratrol daily. Thus, it is highly unlikely that resveratrol is the major contributor to any of the beneficial effects seen with MedDiet consumption. This concern also extends to Section 3 which discusses mechanisms. At the least, the authors need to make clear whether the studies that they cite were done in cells, animals or humans and how the concentrations of resveratrol or oleuropein used in the studies compare to the levels that would be obtained with daily consumption of a MedDiet. As the authors note, EVOO contains multiple polyphenols so it is not clear why they focus on oleuropein, especially as it appears that even most of the mechanistic studies have used mixtures. Thus, the authors need to much better justify their focus on these two polyphenols or else modify the review to reduce that focus. There are also a number of additional points that need to be addressed as listed below.
- Provide a Table with a full description of the different diets discussed in the review. This would be very helpful for readers as not all may know what distinguishes the MedDiet from the other diets that are discussed (eg MIND).
- Sections 3.3 and 3.4 repeat some of the same material. Please remove the repeated material.
- Section 4.2, 1st paragraph: All of these statements need references.
- Section 5.3, 1st paragraph, last sentence: This statement is not clear. What ethnicities showed the most benefits?
- Section 6: The authors need to explicitly state how the levels of resveratrol tested in these studies compare to the levels that an individual would consume on a daily basis when following the MedDiet.
- Section 6: The FDGP used in the Lee et al. study has very low levels of resveratrol (~1.5 mg/kg powder) so it should not be in this Table.
- Section 7, 2nd paragraph, last line: ROO does not have oleuropein so how do the authors explain these results?
- Section 7.2, last line: I thought that the MedDiet was already polyphenol rich and also contained olive oil so I am not clear why adding more would be expected to provide further benefits. Please explain.
- Table 6: Wright et al is not directly relevant to this review as it only looked at diet quality not the MedDiet specifically.
- 26, 2nd paragraph, lines 3-6: The Lee et all study used freeze dried grape powder not grape polyphenols. Please correct.
Author Response
Dear Reviewer,
We would like to express our sincere gratitude for your thorough, detailed, and constructive comments. The manuscript has been revised based on your suggestions, and the key modifications are summarized below:
We have included a summary figure that visually illustrates the neuroprotective mechanisms of the Mediterranean diet.
Tables have been harmonized, duplicate entries removed, and non-relevant studies (e.g Wright et al.) excluded. Additionally, a new table comparing the MedDiet, DASH, MIND, and Western dietary patterns has been incorporated.
The interpretation of the study by Lee et al. has been clarified, and it was removed from the table as a polyphenol intervention, as the study used freeze-dried grape powder containing only minimal resveratrol.
Sections 4.2 and 5.3 have been revised with updated references for all statements, and we have clearly indicated the ethnic groups in which the greatest benefits were observed.
The explanation of the added value of polyphenol supplementation has been expanded, highlighting interindividual variability and differences in bioavailability.
A new, detailed explanatory paragraph has been added to emphasize the low dietary intake levels, dose discrepancies, and differences between in vitro/animal and human studies. We also noted that the polyphenol profile of extra-virgin olive oil is more complex, with the highlighted molecules provided as illustrative examples.
Redundancies in Sections 3.3 and 3.4 have been removed, with clarification that certain biological processes naturally overlap.
The comment regarding the oleuropein-free status of refined olive oil has been addressed, with clarification in Section 7 that the observed effects are likely not attributable to polyphenols.
For improved readability, we have applied more concise formulations throughout the manuscript.
The Discussion and Limitations sections have been further refined and expanded.
We are deeply grateful for your valuable feedback, which has substantially contributed to improving the quality of the manuscript.
Sincerely,
Prof. János Tamás Varga and Co-authors
Round 2
Reviewer 2 Report
Comments and Suggestions for Authors
The authors have done an excellent job at addressing the concerns and questions that I raised in my initial review. Their review is significantly improved and now makes a very useful contribution to the scientific literature.